# CYFRA 21-1 in Lymph Node Fine Needle Aspiration Washout Improves Diagnostic Accuracy for Metastatic Lymph Nodes of Differentiated Thyroid Cancer

**DOI:** 10.3390/cancers11040487

**Published:** 2019-04-05

**Authors:** Jeongmin Lee, Hye Lim Park, Chan-Wook Jeong, Jeonghoon Ha, Kwanhoon Jo, Min-Hee Kim, Jeong-Sun Han, Sohee Lee, Jaseong Bae, Chan Kwon Jung, So Lyung Jung, Moo Il Kang, Dong-Jun Lim

**Affiliations:** 1Division of Endocrinology and Metabolism, Department of Internal Medicine, Seoul St. Mary’s Hospital, College of Medicine, The Catholic University of Korea, Seoul KS013, Korea; 082mdk45@catholic.ac.kr (J.L.); hajhoon@catholic.ac.kr (J.H.); benedict@catholic.ac.kr (M.-H.K.); winehan@catholic.ac.kr (J.-S.H.); mikang@catholic.ac.kr (M.I.K.); 2Division of Nuclear Medicine, Department of Radiology, Seoul St. Mary’s Hospital, College of Medicine, The Catholic University of Korea, Seoul KS013, Korea; prhlim@gmail.com (H.L.P.); ccw2414@naver.com (C.-W.J.); 3Division of Endocrinology and Metabolism, Department of Internal Medicine, Incheon St. Mary’s Hospital, College of Medicine, The Catholic University of Korea, Seoul KS013, Korea; lovi@catholic.ac.kr; 4Department of Surgery, Seoul St. Mary’s Hospital, College of Medicine, The Catholic University of Korea, Seoul KS013, Korea; leesohee@catholic.ac.kr (S.L.); jaseong@gmail.com (J.B.); 5Department of Hospital Pathology, Seoul St. Mary’s Hospital, College of Medicine, The Catholic University of Korea, Seoul KS013, Korea; ckjung@catholic.ac.kr; 6Department of Hospital Radiology, Seoul St. Mary’s Hospital, College of Medicine, The Catholic University of Korea, Seoul KS013, Korea; sljung1@catholic.ac.kr

**Keywords:** fine-needle aspiration, thyroid neoplasms, lymph nodes, CYFRA 21.1, biomarkers

## Abstract

Fine needle aspiration cytology (FNAC) and washout thyroglobulin (Tg) measurements are the standard for evaluating a metastatic lymph node (LN) in thyroid cancer. However, patients rarely benefit from these procedures due to false results. This study aims to identify a reliable biomarker that significantly improves the diagnosis of metastatic LNs, in addition to FNAC and washout Tg. This study analyzed 130 LNs that were suspected to have metastases on thyroid ultrasonography, from June 2016 to December 2017. All subjects underwent FNAC, washout Tg measurements and a new biomarker, washout Cytokeratin fragment 21-1 (CYFRA 21-1) measurement. The final LN outcomes were confirmed by surgical histology, repeat FNAC, or follow-up image. The diagnostic values of the presence of washout CYFRA 21-1 for diagnosing metastatic LNs were evaluated according to final LN outcomes. Among the 130 LNs, 42 were metastatic lesions and 88 were benign. The washout CYFRA 21-1 levels were significantly higher in metastatic LNs than in benign LNs. In contrast to the findings of washout Tg, washout CYFRA 21-1 showed little overlap between benign and malignant LNs, and its diagnostic cutoff values were not affected by surgery. The combinations of FNAC and washout CYFRA 21-1 showed higher sensitivity (91.9%), specificity (96.5%), negative predictive value (98.8%), and diagnostic accuracy (94.2%) than FNAC with washout Tg. The combination of FNAC, washout Tg, and washout CYFRA 21-1 showed the best sensitivity (98.8%). When washout CYFRA 21-1 was applied to the discordant results that were observed between FNAC and washout Tg, 20 of 22 LNs were correctly diagnosed. Washout CYFRA 21-1 measurements in thyroid LNs provide a diagnostic modality.

## 1. Introduction

The incidences of thyroid cancer have increased over the past several decades [1]. Even though thyroid cancer presents indolent behavior and is associated with a good prognosis, lymph node (LN) metastases occurs in 20–31% of patients at the time of thyroid cancer diagnosis and local recurrences occurs in 5–20% of patients during cancer surveillance after initial treatment [2,3]. Fine needle aspiration cytology (FNAC) is useful for the diagnosis of cervical LN metastases because of its high specificity. However, the small size of LNs, cystic changes, and technical method issues cause doctors to miss the presence of scattered or few cancer cells in cytology samples and lead to increased false negative rates, which typically range from 6–8% [4]. Therefore, a supplementary tool should be developed to overcome these issues. 

Thyroglobulin (Tg) is a 660 kilodalton glycoprotein that is produced by thyroid follicular cells, and its appearance in nonthyroidal tissue is an evidence of recurrence or metastasis [5]. The direct measurement of Tg in fine-needle aspiration (FNA) washout fluid (washout Tg) has been suggested to increase the sensitivity of FNAC. Previous studies have reported that washout Tg is more sensitive than FNAC alone, and its combination with FNAC can improve diagnostic accuracy and sensitivity [6,7,8,9,10,11]. 

However, there have been some difficulties associated with the interpretation of washout Tg values in thyroid cancer. First, a wide range of Tg cut-off values have been suggested and lack consensus between several centers [12,13,14]. Second, Tg might be released into LNs due to the presence of normal thyroid tissue before surgery. In this case, high washout Tg levels can be incidentally detected, which could induce false positive results. Third, the normal contralateral lobe may result in false positivity after a lobectomy because the Tg that is released from normal thyroid tissues can be mixed with LN aspiration fluid. The scenarios mentioned above reveal the need for additional biomarkers to confirm LN metastases from thyroid cancer.

Recent studies have measured tumor markers in LNs and FNA washout fluids from various cancers. Cytokeratin fragment 21-1 (CYFRA 21-1) is a soluble fragment of cytokeratin (CK) 19 that has been suggested to be a reliable biomarker due to its high expression in cancer fluids [15]. Several studies have demonstrated that the combination of FNAC and CYFRA 21-1 in body fluids such as lung cancer malignant pleural effusions [16,17] or axillary LN specimens of breast cancer present better diagnostic performance than FNAC alone [18]. 

With respect to thyroid cancer, cystic fluid from thyroid cancer has shown higher concentrations of CK 19 than fluid from benign thyroid tissues [19]. However, no study has investigated the utility of CYFRA 21-1 in the FNA washout fluid of LNs as a complementary tool for diagnosing LN metastases. In this study, washout CYFRA 21-1 in suspicious LNs of thyroid cancer was validated to improve diagnosis of metastatic LNs.

## 2. Results

### 2.1. Clinical Characteristics of Malignant and Benign LNs

One hundred and three patients with 130 LNs were included in this study. The median age of the 103 patients at the time of FNAC was 47.0 years (range 16–83), and 66 (64.1%) were female. The clinical characteristics between benign and metastatic LNs are summarized in Table 1. With respect to the final LN outcomes, 88 (67.7%) LNs were benign and the remaining 42 (32.3%) were metastatic. Among the 88 benign LNs, FNACs of 37 LNs (42.0%) were performed at that time of initial diagnosis, and FNACs of 51 LNs (54.6%) were performed during the follow-up period, after total thyroidectomy or lobectomy. In 42 metastatic LNs, FNAC of 22 LNs (52.4%) were performed at the preoperative period, and FNACs of 20 LNs (47.6%) were performed after total thyroidectomy or lobectomy. Eighty of the 130 LNs were histologically confirmed after LN resection, and the final outcomes of 50 LNs were diagnosed using follow-up images or repeat FNACs. The 42 metastatic LNs originated from 40 papillary thyroid carcinomas and two anaplastic thyroid carcinomas. There were no significant differences in serum Tg and Tg antibody (Ab) levels between benign and metastatic LNs. Washout Tg and washout CYFRA 21-1 presented higher concentrations in metastatic LNs than benign LNs, with statistically significance (*p* = 0.000 and *p* = 0.000, respectively). While the washout Tg values showed significant overlap between benign and malignant LNs, the CYFRA 21-1 values had little overlap between benign and malignant LNs (Figure 1).

### 2.2. The Cut-Off Values of Washout Tg and Washout CYFRA 21-1

In our analysis of the diagnostic performance of washout Tg and washout CYFRA 21-1, metastases were determined using a cutoff value of 11.6 ng/mL (area under the curve (AUC) 0.908, sensitivity 92.3%, and specificity 85.7%) for washout Tg and 1.1 ng/mL (AUC 0.929, sensitivity 92.3%, and specificity 91.2%) for washout CYFRA 21-1, respectively (Figure 2). The cut-off value of preoperative washout Tg was 10.0 ng/mL (AUC 0.919, sensitivity 90.9%, and specificity 87.5%, 95% confidence interval (CI) 0.84–0.994) and the postoperative washout Tg was 114.4 ng/mL (AUC 0.924, sensitivity 80%, and specificity 84.3%, 95% CI 0.865–0.982). However, there were no significant differences of cutoff values in washout CYFRA 21-1 between preoperative and postoperative LNs; the preoperative cut-off value for washout CYFRA 21-1 was 0.85 ng/mL (AUC 0.965, sensitivity 90.9%, and specificity 99.5%, 95% CI 0.904–0.990) and the postoperative cutoff value of washout CYFRA 21-1 was 0.98 ng/mL (AUC 0.967, sensitivity 95%, and specificity 98%, 95%, CI 0.918–0.999).

### 2.3. Individual Diagnostic Performance of FNAC, Washout Tg, and Washout CYFRA 21-1, and Their Correlations with Final Outcomes

Table 2 demonstrates the correlation of FNAC, washout Tg, and CYFRA 21-1 with final outcomes. Among the final outcomes of 42 metastatic LNs, FNAC missed six, but both washout Tg and washout CYFRA 21-1 missed only three. Of the 88 LNs of which the final outcomes were confirmed as benign, washout Tg misclassified 15 as metastatic LNs, while FNAC and washout CYFRA 21-1 misclassified only three and two, respectively.

### 2.4. Comparison of the Diagnostic Performances of FNAC, Washout Tg, Washout CYFRA 21-1, and Their Combinations

Next, the diagnostic performance of FNAC, washout Tg, washout CYFRA 21-1, and their combinations for predicting metastatic LNs were evaluated (Table 3). FNAC showed lower sensitivity and higher specificity than washout Tg and washout CYFRA. Washout CYFRA 21-1 showed better specificity and NPV than washout Tg, leading to the best diagnostic accuracy. The combination of FNAC and washout Tg showed improved sensitivity (88.1%) compared to FNAC alone. CYFRA 21-1 alone showed high sensitivity, specificity, PPV, NPV, and diagnostic accuracy compared to FNAC or washout Tg alone. Adding washout CYFRA 21-1 to FNAC and washout Tg resulted in the highest sensitivity and PPV, although the overall diagnostic accuracy deceased.

### 2.5. CYFRA 21-1 Diagnostic Performance in Discordant LNs between FNAC and Washout Tg

We evaluated the diagnostic performance of washout CYFRA 21-1 in discordant cases, between FNAC and washout Tg. In total, 108 of 130 LNs (83.1%) showed concordant results between FNAC and washout Tg (Figure 3). FNAC and washout Tg results were discordant in 22 of the 103 LNs (16.9%) (eight metastatic LNs and 14 benign LNs). Six of the eight metastatic LNs (seven PTCs and one anaplastic carcinoma) with positive washout Tg (cut-off value 11.1 ng/mL) showed negative in FNAC (five negative for malignancy and one cystic fluid only) (Table 4). Among 14 confirmed benign LNs, washout Tg showed positivity in 13 LNs and FNAC showed positivity (one suspicious metastatic cancer) in only one of the 14 LNs. When washout CYFRA 21-1 was applied to these discordant samples, 90.9% of LNs (20/22) (eight true positive and 12 true negative) were correctly diagnosed in this study.

## 3. Discussion

The measurement of FNA washout Tg in thyroid cancer was proposed in 1992 by Pacini et al. [10], as a supplementary tool to FNAC for the detection of cervical LNs metastases. Several studies have reported that washout Tg was more sensitive than FNAC for detecting metastases and that the sensitivity of FNAC improved after combination with washout Tg [5,20,21,22,23]. The sensitivity (92.9%) of washout Tg was higher than that of FNAC alone (85.7%) and sensitivity was increased from 85.7% to 88.1% when combined with washout Tg. Despite this improvement in diagnostic sensitivity, a diagnostic cut-off value has not been established, and interference from increased serum TgAb levels may induce false negative results [24]. Moreover, as Tg can be produced by normal thyroid tissues, preoperative status, remnant tissue after surgery and/or remnant ablation, and lobectomy status may decrease the diagnostic accuracy of washout Tg for detecting metastatic LNs. To overcome these problems, we seek new tumor markers that can improve the diagnosis of metastatic LNs.

Some proteins are not directly expressed by the tumor but can still be candidates for tumor detection because they can reflect reactions such as thrombocyte activation and inflammation that are closely related to carcinogenesis [25]. Therefore, to seek the biomarker that directly reflects the characteristics of the metastases in the lymph nodes is crucial.

Some studies have found that tumor marker concentrations in body fluids from metastatic cells present higher serum marker concentrations, and the measurement of tumor markers in body fluids or FNA washouts might facilitate the diagnosis of metastases [26,27,28]. CK 19 is a member of the cytokeratin family that is released into the extracellular space during cell cycle epithelial differentiation, tumor cell mortality, and apoptosis [29]. CYFRA 21-1, which is a partial fragment of CK 19, can be detected in serum and body fluids by the CYFRA 21-1 antibody [30]. Elevated CYFRA 21-1 serum concentrations make this protein a reliable tumor marker for the diagnosis of lung cancer progression [31]. In breast cancer, the measurement of FNA washout CYFRA 21-1 in axillary LNs has supported the diagnosis of metastases [32,33].

CK 19 is highly expressed in differentiated thyroid carcinoma, especially in papillary subtype [34]. CK 19 immunostaining may be helpful in thyroid cancer diagnosis as a supplement to the classical cytological diagnosis between thyroid cancer and benign thyroid nodules [35]. The serum CK 19 fragment (CYFRA 21-1) stands out as a circulating marker in dedifferentiated thyroid cancers such as anaplastic thyroid carcinoma and poorly differentiated carcinoma [36,37]. CYFRA 21-1 expression has been investigated in blood and in papillary thyroid carcinoma cystic fluid. One recent study suggested that the high expression of CK 19 in FNAC facilitates the preoperative diagnosis of cystic thyroid lesions [19]. Based on these results, it could be possible to measure CYFRA 21-1 levels in the FNA washout of LNs.

In this study, we aimed to determine whether washout CYFRA 21-1 levels, in addition to FNAC and washout Tg, may be useful for diagnosing metastatic LNs in thyroid cancer prior to thyroidectomy or at follow-up after thyroidectomy. Washout CYFRA 21-1 levels were measured in the remaining sample after FNAC and washout Tg for LNs that was suspicious on thyroid ultrasound (US). A various range of washout Tg cut-off values, from 0.2 to 50 ng/mL, have been suggested in previous studies [4,11,12,21,22,23,25]. The best cut-off value of washout Tg was 11.6 ng/mL (sensitivity 92.3% and specificity 85.7%), and the best cut-off value of washout CYFRA 21-1 1.1 ng/mL (sensitivity 92.3% and specificity 91.2%). While different washout Tg cutoff values have been used based on thyroid operation status, washout CYFRA 21-1 levels showed consistent cutoff values regardless of thyroidectomy status.

Tg is produced in thyroid cancer, benign tumors, and normal thyroid tissue. Therefore, when both malignant and benign or normal tissues exist simultaneously, we cannot know whether increased Tg levels originate from the malignant or benign tissues [38]. In the clinical setting of patients with initial diagnoses of thyroid cancer who are being evaluated for suspicious LNs before surgery, who had contralateral remnant lesions after lobectomy, or who are candidates for the evaluation of recurrent LNs with unrecognized remnant tissue during cancer surveillance after initial treatment, only washout Tg measurement for the diagnosis of metastatic LNs may miss several suspicious cases, when FNAC results are incomplete. In this sense, washout CYFRA 21-1 levels may be used as a complementary biomarker to support the diagnostic performance of conventional tools such as FNAC and washout Tg.

From our results, washout CYFRA 21-1 can be applied as a second biomarker to clarify discordant results between FNAC and washout Tg. Out of 14 discrepant results of benign LNs, washout CYFRA 21-1 criteria missed only two LNs with false-positive results. The washout CYFRA 21-1 values of these two cases were 1.20 ng/mL and 1.39 ng/mL, respectively, and washout Tg values were 653.36 ng/mL and 317.78 ng/mL, respectively. Due to the low washout CYFRA 21-1 (1.1 ng/mL) cut-off values, these two cases were regarded as positive results, leading to false positives. In total 22 (16.9%) of the 130 LNs could be misdiagnosed by FNAC and washout Tg. By applying washout CYFRA 21-1 criteria, 90.9% (20/22) of patients with incorrect diagnoses could be rescued as an accurate diagnosis. Among the 74 concordant results between FNAC and washout Tg in benign LNs, two LNs were presented as metastatic in FNACs. These two LNs in one patient were dissected during total thyroidectomy. In left level three, these LNs were revealed to be benign LNs in histopathologic results. The final outcome of these LNs proved to be benign in the same LN compartments. In these cases, FNAC was performed prior to thyroidectomy. Therefore, washout Tg levels were high, and FNAC might be performed near the primary thyroid carcinoma. The washout CYFRA 21-1 level of these false positive FNACs and the washout Tg levels were 0.31 ng/mL and 0.82 ng/mL, respectively.

The present study has several limitations. First, some LNs that were defined as benign in the final outcome might not truly be benign, because all benign LNs were not based on postoperative pathology data but rather on clinical follow-ups or surveillance by imaging modalities over a limited time (at least one year). Second, this study has a small number of subjects who contributed to our FNAC and washout Tg results for detecting suspicious LNs. Nevertheless, CYFRA 21-1 showed the best diagnostic accuracy under all of the clinical scenarios that were evaluated in this study. Therefore, further studies with large cohorts are necessary to validate our results. Despite these limitations, to the best of our knowledge, this is the first study to investigate a new reliable biomarker for use in evaluating FNA washout fluid from suspicious LNs, in addition to FNAC and FNA washout Tg.

## 4. Materials and Methods

### 4.1. Study Subjects

From June 2016 to December 2017, 244 patients with thyroid cancer had both FNACs and washout Tg measurements on 338 suspicious LNs at that time of initial diagnosis or during cancer surveillance after surgery and/or remnant iodine ablation. Among these patients, 103 with 130 LNs were enrolled into this study who had samples available after both FNAC and washout Tg measurements. This study complied with the ethical standards of the Helsinki Declaration and was approved by the Catholic University of Korea, Catholic Medical Center, Seoul St, Mary’s Hospital Institutional Review Board (IRB approval No. KC17TESI0328). Written informed consent explaining the purpose and procedures has been obtained from all eligible subjects by clinician.

### 4.2. Tg, TgAb, and Washout Tg Measurements

Tg was measured using a monoclonal antibody immunoradiometric assay (Cisbio Bioassays, Codolet, France). The analytical sensitivity, which indicates the lowest detectable concentration with a probability of 95%, was 0.2 ng/mL. The functional sensitivity at which the efficiency variation was equal to 20% was 0.7 ng/mL. Serum TgAb was measured using a competitive radioimmunoassay kit (ZenTech, Angleur, Belgium) with a functional sensitivity of <15 IU/mL.

### 4.3. US and FNAC of Suspicious LNs

US-guided FNA of suspicious LNs was performed by one of several radiologists. Thyroid and neck US were performed with an HDI 3000 scanner (Advanced Technology Laboratories, Bothell, WA, USA) and an HDI 5000 diagnostic sonography system (Philips Medical Systems, Bothell, WA, USA) with a CL10-5 MHz compact linear array transducer. The criteria and technique for FNACs of suspicious LNs were based on the consensus statement of the Korean Society of Thyroid Radiology [39]. Suspicious LNs were defined as LNs with cystic change, calcification, hyperechogenicity, abnormal vascularity, and loss of central hilar echogenicity on thyroid US. After aspiration, samples were immediately smeared on slides, fixed in 95% ethanol, and processed by both hematoxylin and eosin and Papanicolaou stains. As described in a previous study 25, the same needle and syringe were rinsed with 2 mL of normal saline, and the washout fluid was submitted for washout Tg measurement.

### 4.4. Washout CYFRA 21-1 Measurement

Washout CYFRA 21-1 concentrate was measured in the samples that remained after washout Tg measurements were complete with an immunoradiometric assay kit (ELISA-CYFRA, Cisbio Bioassay, Codolet, France), per manufacturer’s instructions. The washout sample was centrifuged at 3200 revolutions per minute (rpm) for 10 minutes. 100 µL of supernatant and 300 µL of I-125 anti-CYFRA 21-1 were added to the ELSA tube. After gentle mixing with a Vortex-type mixer, the tube was incubated for 20 h at 2–8 °C. The tube contents were aspirated, and 2 mL of washing solution was added, after which the tube was re-emptied. This washing process was repeated three times. Finally, the remaining radioactivity that was bound to the ELISA was measured with a Gamma counter (Packard II E5010 cobra Quantum Gamma Counter, Perkin-Elmer, Waltham, MA, USA). The detection limit was determined to be 0.05 ng/mL, with a probability of 95%. The reference range of CYFRA 21-1 was 0.0–3.6 ng/mL.

### 4.5. FNAC and Final LN Outcomes

The FNAC was interpreted by experienced pathologists who specialized in thyroid cytology. The cytology results were categorized into two groups, and metastatic LNs were defined based on documented metastases from thyroid carcinoma or the presence of highly suspicious atypical cells. Benign LNs were based on documented negativity for malignancy, reactive hyperplasia, or specific LN findings such as Kikuchi’s disease, or blood.

The final LN outcomes were based on postoperative pathologies or FNAC results. LNs were confirmed as metastatic if there was a post-operative metastasis of pathology in the same compartment that was previously noted in FNAC [21,22]. If LN dissection was not performed, LNs were confirmed as metastasis if they presented definite metastatic cytologies with high washout Tg levels and metastatic lesions were identified on neck computed tomography (CT) or magnetic resonance imaging (MRI). Otherwise, the reproducibility of the FNAC results were investigated. LNs were considered as benign if the final post-operative histology of the suspicious LNs confirmed a benignity or if follow-up imaging studies, including thyroid US, did not show any increased size of the LNs in patients who did not undergo surgery [40].

### 4.6. Statistical Analysis

All statistical analyses were performed using SPSS software, version 14 (Chicago, IL, USA). Continuous variables were presented as the mean and standard deviation values if the distributions were normal or as medians if the variables were non-normally distributed. Comparisons of basic clinical characteristics between benign LNs and metastatic LNs were performed by independent t-tests for continuous variables. A receiver operating characteristic (ROC) curve analysis was used to confirm the cutoff levels of washout Tg and washout CYFRA 21-1 in our data. The diagnostic performances of FNAC, washout Tg, and washout CYFRA 21-1 were evaluated with respect to sensitivity, specificity, positive predictive value (PPV), negative predictive value (NPV), and diagnostic accuracy. McNamara’s test was used to perform a statistical comparison of the sensitivity and specificity between the diagnostic tools. All *p*-values < 0.05 were considered significant, and two-sided tests were used.

## 5. Conclusions

In conclusion, washout CYFRA 21-1 can improve the diagnostic accuracy of FNAC and washout Tg in metastatic LNs of thyroid cancer, regardless of thyroid surgery status. Considering its high diagnostic performance, washout CYFRA 21-1 should be evaluated when FNAC and washout Tg show discordant results.

## 6. Patents

We applied for a domestic patent in February 2019 under the title “Method for Diagnosing Lymph Node Metastasis in Thyroid Carcinoma Using CYFRA 21-1” (application number: 10-2019-0021927).

## Figures and Tables

**Figure 1 cancers-11-00487-f001:**
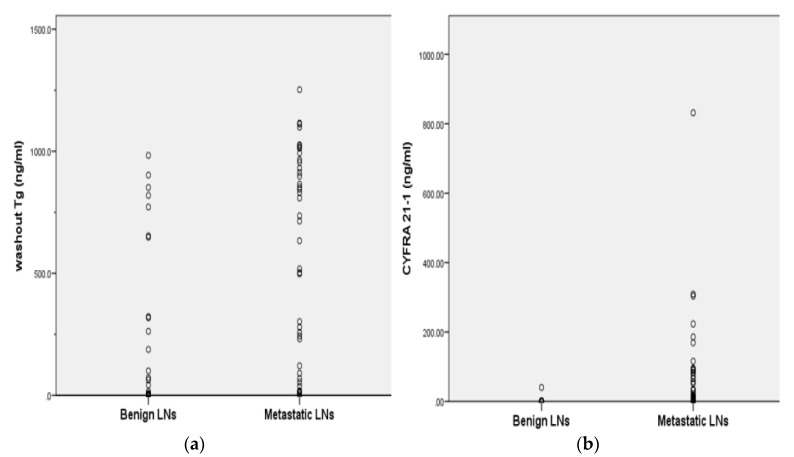
Distribution of washout thyroglobulin (Tg) and washout Cytokeratin fragment 21-1 (CYFRA 21-1) values in benign and metastatic lymph nodes (LNs) on fine-needle aspiration (**a**) Metastatic LNs had a significantly higher median washout Tg than that of benign LNs (*p* = 0.000, by Mann–Whitney U test). The median washout of metastatic LNs and benign LNs was 568.0 and 80.1 ng/mL. (**b**) Metastatic LNs had a significantly higher median washout CYFRA 21-1 than that of benign LNs (*p* = 0.000, by Mann–Whitney U test). The median CYFRA 21-1 of metastatic LNs and benign LNs was 74.2 and 0.9 ng/mL.

**Figure 2 cancers-11-00487-f002:**
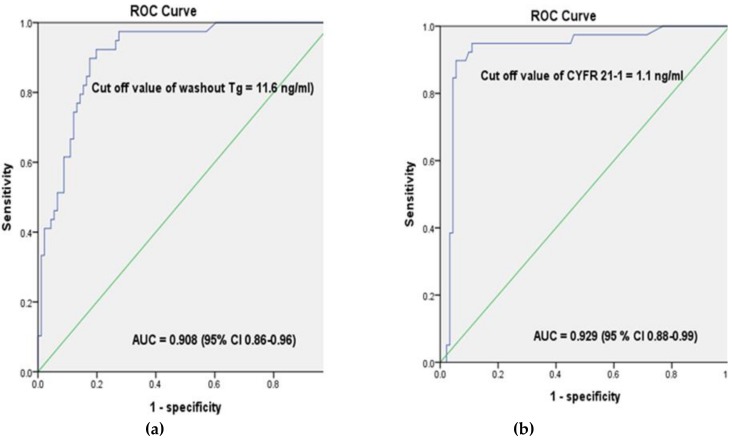
Receiver operating characteristic (ROC) analysis of washout Tg(left) and washout CYFRA 21-1 (right) tested in lymph node fine-needle aspiration (**a**) Metastases were determined using a cutoff value of 11.6 ng/mL (AUC 0.908, sensitivity 92.3%, and specificity 85.7%) for washout Tg and (**b**) 1.1 ng/mL (AUC 0.929, sensitivity 92.3%, and specificity 91.2%) for washout CYFRA 21-1, respectively.

**Figure 3 cancers-11-00487-f003:**
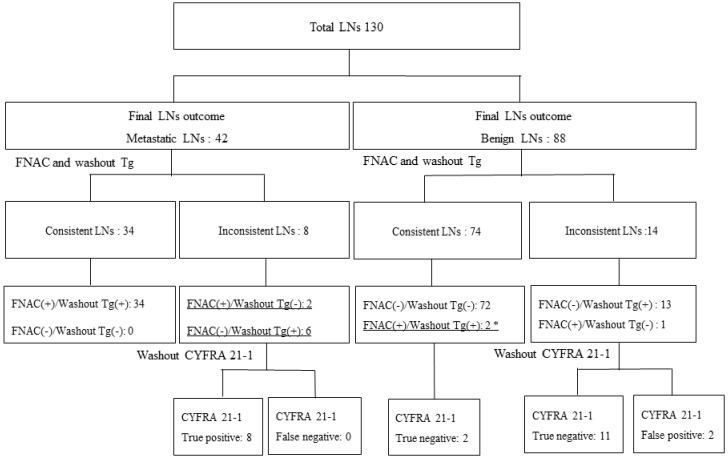
Discordant cases between FNAC and washout Tg 108 of 130 LNs (83.1%) showed concordant results between FNAC and washout Tg. If washout CYFRA 21-1 was applied to these discordant samples, 90.9% of LNs (20/22) (eight true positive and 12 true negative) were correctly diagnosed. * These two LNs within one patient showed highly suspicious FNAC result and high washout Tg level but finally turned out to be benign after compartment dissection. LN, lymph node; FNAC, fine needle aspiration cytology; Tg, thyroglobulin.

**Table 1 cancers-11-00487-t001:** Clinical characteristics of 130 lymph nodes evaluated in the study.

Clinical Characteristics	Benign LN (*n* = 88)	Metastatic LN (*n* = 42)	*p*-Value
FNAC according to operation			0.268
FNAC before surgery (%)	37 (42.0)	22 (52.4)	
FNAC at postoperative follow-up	51 (58.0)	20 (47.6)	
Diagnostic methods for final LNs outcomes			0.000
Histological confirm (%)	42 (47.7)	38 (90.5)	
Confirmed by image or repeat FNAC (%)	46 (52.3)	4 (9.5)	
Serum Tg (ng/mL)	15.5 (0.0–205.0)	18.6 (0.1–126.4)	0.614
Serum TgAb (IU/mL)	53.12 (1.6–895.3)	25.8 (0.1–270.7)	0.319
Washout Tg (ng/mL)	80.1 (0.0–983.2)	568.0 (0.1–1252.4)	0.000
Washout CYFRA 21-1(ng/mL)	0.9 (0.1–40.1)	74.2 (0.2–931.6)	0.000

Data are expressed as median (range) or number including percentage. FNAC, fine needle aspiration cytology; Tg, thyroglobulin; TgAb, thyroglobulin antibody.

**Table 2 cancers-11-00487-t002:** Correlation of FNAC, washout Tg and washout CYFRA 21-1 of 130 LNs by final outcomes.

Diagnosis	FNAC		Washout Tg		Washout CYFRA 21-1	
Final diagnosis	Positive (*n* = 39)	Negative (*n* = 91)	Positive (*n* = 54)	Negative (*n* = 76)	Positive (*n* = 41)	Negative (*n* = 89)
Benign (*n* = 88)	3	85	15	73	2	86
Metastatic (*n* = 42)	36	6	39	3	39	3

FNAC, fine needle aspiration cytology. Washout Tg cutoff value for positivity according to ROC curve analysis; 11.6 ng/mL Washout CYFRA 21-1 cutoff value for positivity according to ROC curve analysis; 1.1 ng/mL.

**Table 3 cancers-11-00487-t003:** Comparison of diagnostic performance of FNAC, washout Tg, washout CYFRA 21-1, combined with FNAC and washout Tg, combined with FNAC and washout CYFRA 21-1 and combine with all modality.

Diagnostic Tool	Diagnostic Value				
	Sensitivity (%)	Specificity (%)	PPV (%)	NPV (%)	Accuracy (%)
FNAC	85.7	96.5	93.4	91.2	93.1
Washout Tg	92.9	83.0	72.2	96.1	86.2
FNAC+ washout Tg	88.1	83.0	92.5	81.1	82.3
Washout CYFRA 21-1	92.9	97.7	95.1	96.7	96.2
FNAC+ washout CYFRA 21-1	91.9	96.5	81.6	98.8	94.2
FNAC+ washout Tg + washout CYFRA 21-1	98.8	93.1	98.8	88.2	90.0

FNAC, fine needle aspiration cytology; PPV, positive predictive value; NPV, negative predictive value.

**Table 4 cancers-11-00487-t004:** The clinical characteristics of 22 LNs with discordant results between FNAC and washout Tg.

No.	Gender	Age	Time of FNAC	Operation	Serum Tg	Serum TgAb	FNAC	Washout Tg	Washout CYFRA 21-1	Final LN Outcome
Metastatic LNs										
1	F	82	During follow-up post-thyroidectomy	Total thyroidectomy	0.17	0.10	Normal thyroid remnant	16.67	831.60	Anaplastic carcinoma
2	M	63	During follow-up post-thyroidectomy	Total thyroidectomy	40.67	0.37	Metastatic LN	2.45	1.33	Metastatic PTC
3	M	57	Initial diagnosis	Total thyroidectomy	29.56	3.24	Cystic fluid	633.10	73.30	Metastatic PTC
4	F	40	Initial diagnosis	Total thyroidectomy	0.62	67.57	Negative for malignancy	120.88	33.45	Metastatic PTC
5	M	27	Initial diagnosis	Total thyroidectomy	5.95	6.34	Negative for malignancy	18.37	14.95	Metastatic PTC
6	F	41	Initial diagnosis	Total thyroidectomy	83.12	6.00	Negative for malignancy	993.03	3.02	Metastatic PTC
7	M	38	Initial diagnosis	Total thyroidectomy	11.23	9.72	Negative for malignancy	14.4	1.24	Metastatic PTC
8	F	20	Initial diagnosis	Rt. Lobectomy	7.68	7.41	Metastatic LN	1.33	4.00	Metastatic PTC
Benign LNs										
1	F	50	During follow-up post-thyroidectomy	Total thyroidectomy	41.93	6.00	Negative for malignancy	317.78	1.39	No lesion on follow-up neck CT
2	F	69	During follow-up post-thyroidectomy	Total thyroidectomy	0.21	8.62	Metastatic cancer	0.16	1.20	Blood only Decreased in size on neck CT
3	M	32	During follow-up post-thyroidectomy	Total thyroidectomy	0.20	9.02	Negative for malignancy	771.83	0.96	Negative for malignancy
4	F	51	During follow-up post-thyroidectomy	Total thyroidectomy	0.09	8.83	Cystic fluid only	902.17	0.75	Negative for malignancy
5	F	82	During follow-up post-thyroidectomy	Total thyroidectomy	0.17	0.10	Negative for malignancy	819.23	0.66	Negative for malignancy
6	M	39	During follow-up post-thyroidectomy	Total thyroidectomy	0.20	14.00	Negative for malignancy	42.28	0.43	Negative for malignancy
7	F	57	During follow-up post-thyroidectomy	Total thyroidectomy	1.74	8.03	Negative for malignancy	100.22	0.29	Negative for malignancy
8	M	47	During follow-up post-thyroidectomy	Rt. Lobectomy	0.19	91.76	Negative for malignancy	188.14	0.34	Negative for malignancy
9	F	37	During follow-up post-thyroidectomy	Rt. Lobectomy	7.57	52.31	Negative for malignancy	15.70	0.19	Negative for malignancy
10	M	44	During follow-up post-thyroidectomy	Lt. Lobectomy	2.27	183.79	Negative for malignancy	653.36	0.96	Negative for malignancy
11	M	44	During follow-up post-thyroidectomy	Lt. Lobectomy	2.27	183.79	Negative for malignancy	262.53	0.71	Negative for malignancy
12	M	44	During follow-up post-thyroidectomy	Lt. Lobectomy	2.27	183.79	Negative for malignancy	648.03	0.35	Negative for malignancy
13	F	46	Initial diagnosis	Total thyroidectomy	41.93	7.74	Blood only	63.21	0.52	Negative for malignancy
14	M	71	Initial diagnosis	Total thyroidectomy	37.37	4.16	Negative for malignancy	71.78	0.29	Negative for malignancy

FNAC, fine needle aspiration cytology; Tg, thyroglobulin; TgAb, thyroglobulin antibody.

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
