# Peer review of "CYFRA 21-1 in Lymph Node Fine Needle Aspiration Washout Improves Diagnostic Accuracy for Metastatic Lymph Nodes of Differentiated Thyroid Cancer"

_cancers, 2019, doi:10.3390/cancers11040487_

Round 1

Reviewer 1 Report

The manuscript deals with the possibility to use a new biomarker, namely the cytokeratin fragment CYFRA 21-1 to improve diagnosis of lymph node (LNs) metastasis in thyroid cancer. The results are promising and show that the use of the new biomarker increases the sensitivity and specificity of identifying LNs as compared to fine needle aspiration cytology (FNAC) alone, and with its combination with thyroglobulin (Tg) washout, decreasing incorrect diagnoses.

Overall, the manuscript is well written and I suggest its publication. I point out only minor revisions.

At line 273 consider to use the singular form of verb “reflect”.

The statement at lines 180-184 appears unclear. Consider to rephrase it.

At line 198 please write full name for US. Then the authors may avoid rewriting full name at line 257.

At line 224, why did the authors write that in the two benign LNs among the concordant cases, they are metastatic only in FNACs? In figure 3 they are also positive in Tg washout.

At line 247, please correct “written”.

Author Response

We appreciate your and reviewers' efforts for reviewing our manuscript. In the revised manuscript, we made corrections according to reviewers' comments and suggestions. A response to the referees΄ suggestions has been listed one by one. Modified contents were written in highlight in the manuscript and we used the "Track Changes" function in Microsoft Word. We made a correction for some typos. We deleted the duplicate reference and reset the reference number.

Reviewer 2 Report

In the paper “CYFRA 21-1 in Lymph Node Fine Needle Aspiration Washout Improves Diagnostic Accuracy for Metastatic Lymph Nodes of Differentiated Thyroid Cancer” Jeongmin Lee and colleagues found that the washout of CYFRA 21-1 in suspicious LNs of thyroid cancers improves their diagnosis as metastatic LNs.

The method was previously developed for other cancer types, thus it represents the first study in the context of thyroid tumors, where circulating CYFRA 21-1  was proposed only as a prognostic and dedifferentiation marker  (Giovanella L et al, 2012 and 2017; Isic T. et al, 2010).

The findings are interesting, clearly presented and innovative; the  results are clearly described and  discussion takes into account strengths and limitations of the study.

However , there are some minor comments as follow:

·   In the text , the authors often misspelled FNAC with FANC (i.e in Figure 3, Table 2);

·    In Figure 1, median values of washout Tg and CYFRA 21-1 are reported , although  slightly different from those reported in Table 1; authors should standardize the data or at least clearly indicate if they refer to median or mean;

· In Paragraph 2.2, lane 116, the authors should state “three and two, respectively”, according to Table 2;

· The Reference 5 (Giovanella L, Diagn Cytopathol 2009, 37, 42-44) was incorrectly mentioned twice (Ref 24).

Author Response

We appreciate your and reviewers' efforts for reviewing our manuscript. In the revised manuscript, we made corrections according to your comments and suggestions. A response to your suggestions has been listed one by one. Modified contents were written in highlight in the manuscript and we used the "Track Changes" function in Microsoft Word. We made a correction for some typos. We deleted the duplicate reference and reset the reference number.
